# A Holistic Approach to Use Educational Robots for Supporting Computer Science Courses



Zhumaniyaz Mamatnabiyev [1], Christos Chronis [2], Iraklis Varlamis [2,*], Yassine Himeur [3,*] and Meirambek Zhaparov [4]

1   Department of Computer Sciences, SDU University, Abylaikhan Street 1/1, Kaskelen 040900, Kazakhstan; zhumaniyaz.mamatnabiyev@sdu.edu.kz
2   Department of Informatics and Telematics, Harokopio University of Athens, Omirou 9, 17778 Athens, Greece
3   Faculty of Engineering and Information Technology, University of Dubai, Academic City, Emirates Road-Exit 49, Dubai P.O. Box 14143, United Arab Emirates
4   ICT Faculty, Paragon International University, Phnom Penh 12151, Cambodia; mzhaparov@paragoniu.edu.kh
*   Correspondence: varlamis@hua.gr (I.V.); yhimeur@ud.ac.ae (Y.H.); Tel.: +30-210-9549405 (I.V.)

**Abstract:** Robots are intelligent machines that are capable of autonomously performing intricate sequences of actions, with their functionality being primarily driven by computer programs and machine learning models. Educational robots are specifically designed and used for teaching and learning purposes and attain the interest of learners in gaining knowledge about science, technology, engineering, arts, and mathematics. Educational robots are widely applied in different fields of primary and secondary education, but their usage in teaching higher education subjects is limited. Even when educational robots are used in tertiary education, the use is sporadic, targets specific courses or subjects, and employs robots with narrow applicability. In this work, we propose a holistic approach to the use of educational robots in tertiary education. We demonstrate how an open source educational robot can be used by colleges, and universities in teaching multiple courses of a computer science curriculum, fostering computational and creative thinking in practice. We rely on an open-source and open design educational robot, called FOSSBot, which contains various IoT technologies for measuring data, processing it, and interacting with the physical world. Grace to its open nature, FOSSBot can be used in preparing the content and supporting learning activities for different subjects such as electronics, computer networks, artificial intelligence, computer vision, etc. To support our claim, we describe a computer science curriculum containing a wide range of computer science courses and explain how each course can be supported by providing indicative activities. The proposed one-year curriculum can be delivered at the postgraduate level, allowing computer science graduates to delve deep into Computer Science subjects. After examining related works that propose the use of robots in academic curricula we detect the gap that still exists for a curriculum that is linked to an educational robot and we present in detail each proposed course, the software libraries that can be employed for each course and the possible extensions to the open robot that will allow to further extend the curriculum with more topics or enhance it with activities. With our work, we show that by incorporating educational robots in higher education we can address this gap and provide a new ledger for boosting tertiary education.

**Keywords:** educational robots; computer science; FOSSBot; computer science curriculum

## 1. Introduction

Robots constitute examples of intelligent technology with applications in many areas such as healthcare, industry, education, etc. During the last decade, they have become a promising technology for reshaping K-12 education, especially STEM (Science, Technology, Engineering, and Math) education [1]. Nowadays, various educational robots are available, with wide functionalities, from child development in kindergartens and primary

schools to supporting algorithmic knowledge gained in high schools, programming, and automation in higher education. For example, BeeBot is used for tutoring preschool and school children to use simple instructions to navigate robots in complex environments and scenarios [2]. Another popular educational robot is Thymio which has been applied in education of different age groups including adults and elderly people who are interested in robots [3]. Both robots have a great impact on STEM (science, technology, engineering, and mathematics) education where they engage learners on development, simulation, and problem-solving processes [4,5]. However, robotics in primary education have become an even more prevalent solution with the advent of LEGO [6,7] that promoted the concept of robotic parts that can be assembled in a multitude of different ways thus fostering student creativity and innovation.

At primary and secondary schools, educational robots are involved in improving students' algorithmic and programming skills [8]. They are involved in learning subjects from sciences such as physics, from engineering such as mechatronics, as well as from algorithms and programming. Smart user interfaces that allow to interact with the robot with visual coding or no-coding, like Scratch (https://scratch.mit.edu, accessed on 16 January 2023), allowed younger students to learn the fundamentals of algorithms and make their first steps in programming at an early age. Competitions among primary and secondary school children attain learners' interest while letting them demonstrate problem-solving and presentation skills by ordering robots to perform various tasks [9].

In educational institutes, educational robots can play a big role in teaching computer science and engineering-related courses like algorithms, programming, physics, mechatronics, electronics, etc. where students create their own robots and test them in real-life cases. Hands-on projects allow students to directly experiment on various tasks, ranging from the design or assembly of the robot to the operation of its hardware and the programming of its software stack. For example, an educational robot can be employed in the assignments of a Machine Learning (ML) and Artificial Intelligence (AI) course that comprise line following, static image recognition, and scene understanding tasks [10]. Such assignments require the use of the microcontroller, the sensors, and actuators of the robot, and allow students to put into practice what they have learned in theory in order to solve tangible problems, thus improving their creativity and motivation to do more tasks and experiment on more complex scenarios.

The existing literature on educational robotics in tertiary education mostly focuses on using commercial robots or their virtual simulations for supporting specific courses and teaching specific subjects individually [11,12]. They usually lack a holistic approach in using the same robot to support all the courses of a curriculum and provide activities that fit the needs of each specific course. The main objective of the current work is to fill this gap by using an open-source robot and providing an approach for using it in multiple courses of a Computer Science curriculum. We believe that an open-source robot allows for more customization and expansions and can be the common basis for a holistic approach in the use of Educational Robots for supporting multiple courses of a curriculum. This provides better integration of emerging technologies -especially robotics- into higher education through actual practice and engagement [13].

Courses designed around open hardware and software robots allow students to get the full capabilities of a robot. For example, the controller of a robot can be involved in programming sensors and actuators through GPIO (general-purpose input/output) pins. The raw data collected from sensors can be used for learning how to analyze, visualize, train, and test when creating artificial intelligence (AI) models. Actuators can be used to execute the actions suggested by the models. Similar processes can be involved in a variety of activities during many courses of computer science curricula.

This work presents a holistic approach to utilizing educational robots in teaching and learning computer science at colleges and universities. An extensive review of educational robots, that are integrated into academic curricula, shows there is a significant gap in the literature—the absence of a curriculum that is fully integrated with an educational robot.

To address this gap, we propose a one-year computer science curriculum that comprises several computer science and engineering courses, which base their practical part and assignments on educational robots. The courses of the curriculum comprise electronics, embedded systems, python programming, algorithms, data science, artificial intelligence, natural language processing, computer vision, cloud computing, cybersecurity, and the Internet of Things. The curriculum takes advantage of the capabilities of educational robots, which are a combination of both hardware and software components and examines various aspects of educational robots from their composition to their use.

An open-source and open-design educational robot, FOSSBot [14], is selected as a tool for learning activities during the courses. FOSSBot allows learners to make changes to its design, to easily attach more sensors or external devices in order to expand its scope of use. It supports various activities and can be used at all levels of education. Another benefit of using FOSSBot is the plastic frame that can be printed in a 3D printer making it cheap. Many different sensors and actuators are available that increase its usage in activities. The software stack of the robot runs on a Raspberry Pi microprocessor. Its flexible stack, which is written in Python, lets users write their programs either in a no-code visual interface, in Python using Jupyter notebooks, or in raw Python scripts that interact directly with the core robot components (i.e., sensors, actuators, and the microprocessor). The onboard Wi-Fi module of Raspberry Pi allows to remotely communicate with FOSSBot, thus setting more challenges for students and their professors that teach them communications and security.

The proposed curriculum takes advantage of FOSSBot's capabilities, allowing students to engage in hands-on activities throughout their learning journey. These activities encompass designing and assembling the robot, operating its hardware, and programming its software stack. The curriculum design considered the software libraries required for each course and explored potential extensions to the FOSSBot platform. This approach aimed to provide a comprehensive and practical learning experience, enabling students to apply theoretical concepts in real-life scenarios and enhance their creativity and motivation. Activities related to the robot components and courses are thoroughly discussed and described within the work. This involves a comprehensive examination of the specific tasks and exercises that engage students in hands-on interactions with FOSSBot and its various components.

The main contributions of the current work can be summarized in the following:

- a generic approach for using educational robots in multiple courses at a tertiary education level, supported by an open-source educational robot,
- a detailed list of computer science courses that can use the educational robot to support their activities,
- a list of software libraries that can be used to expand the python-based software stack of the robot in supporting course-specific activities, and the list of sensors and actuators that are needed in the related activities,
- a list of indicative activities that fit the context of courses, and take advantage of a variety of sensors that are or can be attached to the open-source robot.

The article is structured as follows: Section 2 performs an overview of previous related works and discusses other educational robots and their applications in various levels of education. Section 3 describes the main building block of the proposed approach, FOSSBot, its software and hardware components. Section 4 explains the structure of the proposed curriculum and the courses it covers, the specific libraries that are needed to extend FOSSBot's main software stack in order to support the activities of the courses in the curriculum and the recommended activities that relate to each one of the many FOSSBot sensors. Section 5 performs a two-stage evaluation of the proposed intervention, first in terms of a university course and second in an informal educational context, and reports our results. Section 6 concludes the article by providing future research directions.

## 2. Related Work

Computational thinking has been applied in computer science education for more than a half-century. "It represents skill sets that everyone would be eager to learn and use" and it must be added to learners' analytical ability in reading, writing, and arithmetic, not only computer science, as mentioned by Wing [15] (pp. 1). The concepts of computational thinking are abstraction, algorithm, decomposition, generalization, logical analysis, and evaluation, applied in any other fields [16,17]. Korkmaz et al. [18] proposed a set of Computational Thinking Scales to evaluate students' skill levels in creativity, algorithmic thinking, collaboration, critical thinking, and problem-solving, and then used educational robots to analyze students' computational thinking skills in these scales [19]. According to Zinonos et al. [9], computational thinking is one of the six learning outcomes of Educational Robots together with problem-solving skills, self-efficacy, creativity, motivation, and collaboration. Educational robots improve students' computational thinking skills and STEM attitudes [4,6–8]. Students attain knowledge in various topics starting from how to build a robot to how to interact with it using the sensors and actuators attached to it.

A major part of the educational robot literature is on the use of commercial educational robots. Most of the commercial educational robots are designed for learning programming and algorithmic concepts in different age groups and the existing educational platforms are categorized as no code, basic code, advanced code [9]. For example, BeeBot [2] is used to give basic instructions to kindergarten children and exercise their problem-solving skills, by programming it to navigate through different obstacles or follow certain paths, and at the same time supports basic STEM education with activities that engage children and encourage their creativity. Thymio II [20] helps children to get on board fast using pre-defined behaviors, but also enhances their creativity by supporting robot programming either via the graphical no-code interface or via coding, and LEGO education series [6] proposes several models to work with graphical representation of commands in different ages. The latter is very popular in secondary education and is frequently used in robotics competitions among young students. However, it is not open source and has a high cost, comes with a pre-defined set of compatible sensors and actuators, and thus is less popular in higher education. Another commercial robot, mBot [21] that is based on ATmega328, has the ability to connect external sensors and actuators that are not in the original kit thus providing better expandability. mBot can be programmed using either its visual coding language (mBlocks) or plain code in Python and C. Edison [22] is another well-known robot that is involved in elementary and secondary education, it is cheaper than other commercial robots, but cannot be easily extended with additional sensors and actuators, which reduces its applications.

On the other side, we have educational robots that are the result of research projects, or robots that are freely available as open-source projects. Such examples are Hydra [23], EUROPA [24], and FOSSBot [11], which comprise a multitude of sensors and actuators and have been designed for teaching various courses and supporting different activities. Due to this variety of sensors and actuators, they can be useful tools for teaching various STEM education subjects if the respective courses are properly organized and supported by educational material [25]. Their main differences when compared to commercial robots are their lower cost, the flexibility to fit the needs of different courses through redesign, and their accessibility due to their open nature or the use of open standards [24]. The EUROPA robot [24] has been used in secondary and higher education and is based on a Raspberry Pi microprocessor, camera, and LIDAR sensors in order to help students learn programming and computer vision. The Hydra robot [23] is an Arduino microcontroller-based robot, and uses an ultrasonic sensor and several actuators and the scratch programming language in order to teach children how to program. Duckiebot [26] is another higher education robot that uses the capabilities of the Raspberry Pi microprocessor. It employs the popular Robot Operating System (ROS) and can be programmed in Python. The main disadvantage of Duckiebot is that it only uses cameras as sensors. Unlike Duckiebot, FOSSBot [14] has a large number of sensors and actuators that could be used in teaching different subjects and

can support all educational levels, since it combines a no-code programming interface and Python programming capabilities.

In order to summarize the features of the various educational robots that we found in our study, we provide their main characteristics in Table 1 organized in the sensors and actuators they use, their data communication method, the level of education they support, and the supported programming modes.

**Table 1.** List of sensors of different educational robots compared.

| Robot | Level of Education | Programming Mode | Controller/CPU | Sensors | Actuators | Data Communication |
|---|---|---|---|---|---|---|
| EUROPA [24] | Secondary, Higher | Advanced | Raspberry Pi | Ultrasonic, camera, LIDAR, infrared, optical encoder | DC motor, robotic arm, LED | Wi-Fi |
| Hydra [23] | Secondary, Higher | Basic code | Arduino | Ultrasonic, potentiometer, button | DC motor, LED, seven segment display, RGB LED | Direct |
| LEGO EV3 [6] | Secondary | No code, Basic code | ARM9 | Ultrasonic, touch, color | Gear motor | Wi-Fi, Bluetooth |
| Thymio II [3] | Kindergarten, Primary, Secondary | No code, Basic code, Advanced | n/a | Infrared, accelerometer, microphone, temperature sensor, button | DC motor, speaker, LED | IR |
| mBot [27] | Elementary, Secondary | No code, Basic code, Advanced | ATmega328 | Button, ultrasonic, line following, light sensor, infrared | DC motor, RGB LED, buzzer | Direct |
| Edison [27] | Elementary, Secondary | No code, Basic code, Advanced code | MC9508PA16 | Button, infrared, light, line following, microphone, optical encoder | DC motor, LED, buzzer | IR |
| Duckiebot [26] | Higher | Advanced | Raspberry Pi | Camera | DC motor | Wi-Fi |
| FOSSBot [14] | Kindergarten, Primary, Secondary, Higher | No code, Basic code, Advanced | Raspberry Pi | Accelerometer, gyroscope, odometer, photoresistor, infrared, ultrasonic | RGB LED, DC motor, speaker | Wi-Fi, Bluetooth |

The advantage of using robots in education is that they allow students to learn course materials using hands-on activities. Activities on computer science-related subjects ask learners to solve real-world problems. These advantages have been recognized by universities, which have already involved the use of Lego Mindstorms EV3 robot in computer programming courses, such as the one in [22] for engineering students that Lego Mindstorms EV3 and in [28], the authors engaged middle school students to learn programming concepts using educational robots, using activities such as line following, hand control, touch control, speech control, and body control. Another subject that could be taught using robots is described in [25]. The authors discussed their experiment conducted in a Robotics course using an IoT collaborative-supported learning method. Artificial intelligence, deep learning, and computer vision-related topics are mapped in courses where data from multiple sensors are used to create a model and test it [10,29,30]. Some activities include line

following, mapping, and object recognition. Sensors, actuators, microcontrollers and microprocessors are the base of robot construction, and used for learning embedded systems and electronics-related subjects [31,32].

Although all existing approaches have been designed for selected courses in a computer science program, there is still no organized attempt to use educational robots in the activities of all courses of a CS curriculum. Some of the educational robots can be involved in multiple courses with additional sensors and actuators on them. FOSSBot is an open-source and open-design educational robot that provides a wide range of sensors and actuators, offers WiFi connectivity, and can be easily expanded with more electronics, thus offering many possibilities for supporting a variety of courses [14]. The Raspberry Pi microprocessor on it allows different programming approaches for artificial intelligence, deep learning, computer vision, embedded systems, etc. Learners from different age groups can interact with it through various ways such as 'no code', scratch, and Python programming. This versatility makes FOSSBot an ideal platform for delivering a comprehensive computer programming curriculum at both undergraduate and postgraduate levels. By emphasizing openness and expandability, FOSSBot addresses the limitations of other educational robotics solutions, offering educators and students alike a dynamic and adaptable tool to support diverse learning objectives and activities across the entire computer science curriculum. In the section that follows, we perform a brief overview of FOSSBot and its capabilities, though we suggest readers consult the original article [14] to get more information.

The analysis of the state-of-the-art research and commercial solutions in the field of educational robotics uncovers several challenges. One notable challenge is the diversity of available educational robots, ranging from commercial products with pre-defined features and limitations to open-source projects offering greater flexibility but requiring additional technical expertise. Integrating these robots into existing curricula presents logistical challenges, including the need for specialized training for instructors, adequate technical support, and the development of suitable educational materials. Moreover, ensuring equitable access to educational robots across different institutions and socioeconomic backgrounds is crucial for promoting inclusivity and diversity in computer science education. Additionally, the effectiveness of educational robots in enhancing learning outcomes may vary depending on factors such as student engagement, instructor expertise, and the alignment of educational activities with learning objectives. Addressing these challenges requires a holistic approach that involves collaboration among educators, researchers, industry partners, and policymakers to develop best practices, standards, and resources for integrating educational robots into tertiary education effectively. By acknowledging these challenges and exploring potential solutions, we can ensure that the integration of educational robots in tertiary education is not only impactful but also sustainable and inclusive. Future research and development efforts should focus on addressing these challenges to maximize the benefits of educational robots in promoting active learning, interdisciplinary collaboration, and innovation in computer science education.

## 3. Materials and Methods

In this section, we present the building blocks of our proposed solution and the main methods employed to deliver it to the educational environment. The main component is an open-source educational robot that can be adapted to enhance computer science education across various courses. We begin with a comprehensive examination of the robot's hardware and software architecture, along with its intricate components. The presentation highlights the open and expandable nature of the basic robot which allows it to easily adapt to the needs of various courses. This detailed presentation is of paramount importance, as it provides a roadmap for fellow researchers to replicate and extend our approach to other educational contexts or disciplines.

FOSSBot [14], which stands for Free and Open Source Software Robot is an open design educational robot developed for learning STEM education. The hardware parts of

the robot can be replaced at a low cost in case of fault since they are electronics that can be easily found in the market. Since its frame is built from plastic, it is easy to print in a 3D printer, and even better it is possible to redesign it to support more activities. The upper part of the frame has been designed to allow attaching a LEGO-compatible base and can be modified to allow more types of constructions to be built on top. Figures 1 and 2 illustrate the FOSSBot and its interior and top, respectively. Moving on, FOSSBot contains most of the sensors and actuators found in commonly used educational robots, as has been shown in Table 1.

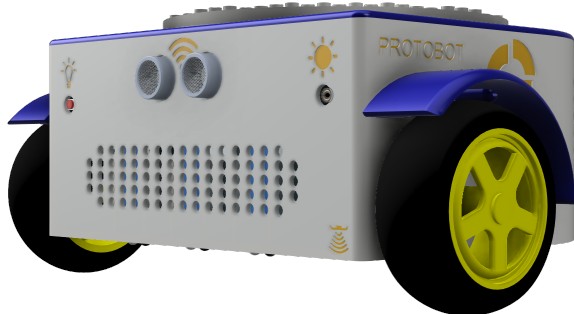

**Figure 1.** FossBot.

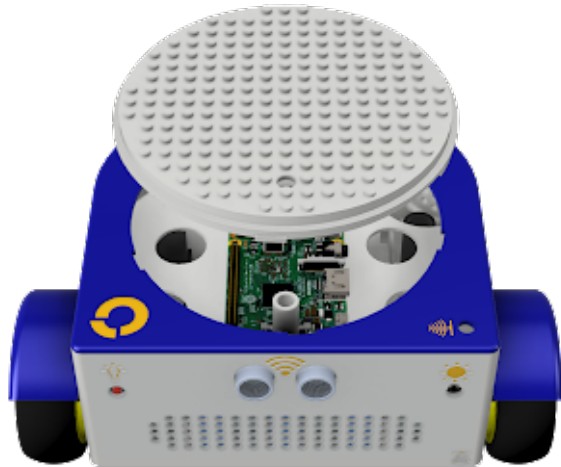

**Figure 2.** FossBot Interior and top.

The robot is based on a Raspberry Pi microprocessor but can be replaced or expanded with other microcontrollers like Arduino, BeagleBone, etc. The microprocessor enables the robot to run in three modes such as no-coding, block-based, and notebook coding modes. The connection to the robot is wireless, using WiFi (see Figure 3). The no-coding mode is suitable for children who are beginning to demonstrate the robot's abilities using a user interface (UI). The block-based coding mode is based on Google Blockly software which is open source. This mode allows learners from primary school to give instructions to the robot, perform different tasks, and interact with the robot's sensors and actuators. Users who are good at coding can program the robot using a Jupyter Notebook in the Python programming language. In addition, experienced users can connect to the robot and use its core library directly using the Python programming language.

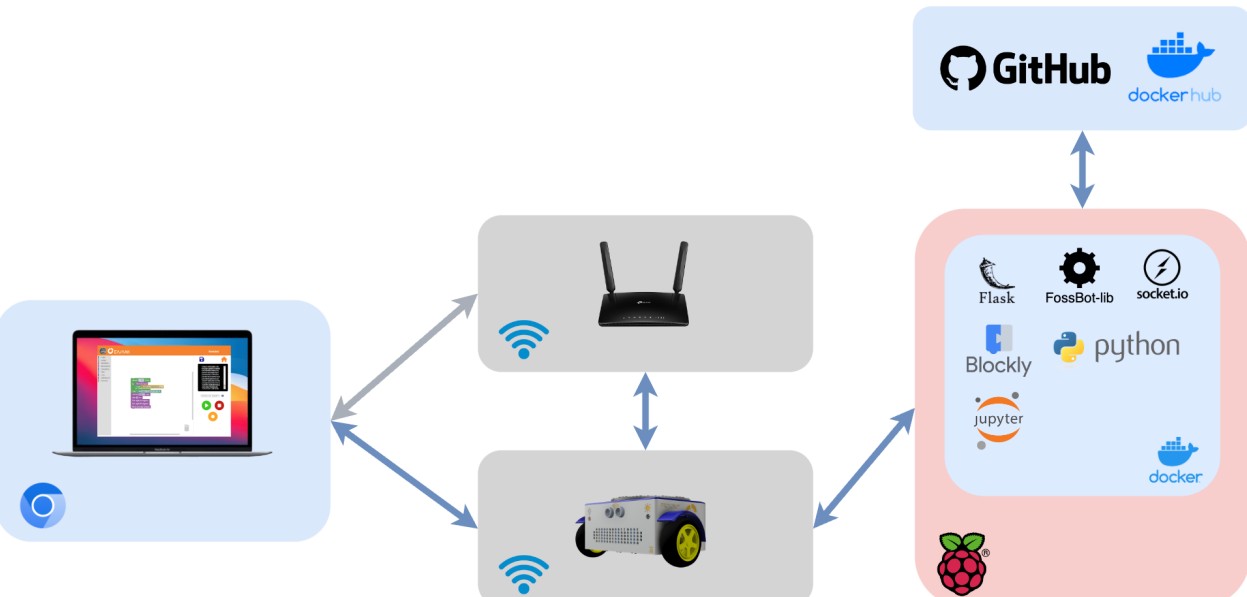

**Figure 3.** FossBot Architecture.

The software of FOSSBot is installed in the microprocessor and it is available on GitHub, thus allowing the open-source community to download and make a contribution to it. The software updates are deployed and integrated continuously with the help of dockerized images (https://github.com/eellak/fossbot, accessed on 2 April 2024).

Compared to other commercial and research robots that are used in educational robotics, as listed in Table 1 and analyzed in [33], FOSSbot offers a rich inventory of sensors at a low cost, nearly 100 euros, which can be further reduced if the battery recharge module is omitted. It is not as popular as LEGO or other commercial robots, but it is as flexible as most research robots, and this flexibility is enhanced by its open-source and open-design nature. This flexibility makes it ideal for a wide range of courses [33].

## 4. Results

The main result of the proposed approach relies upon the unique advantages that robots bring to the educational landscape, not only in electronics but also across diverse domains such as programming, data science, networks, embedded systems, and artificial intelligence. By integrating robots into programming courses, students gain hands-on experience in translating code into real-world applications, reinforcing theoretical concepts through practical implementation. For instance, in data science, students can leverage robots to collect and analyze data, gaining insights into the practical applications of data-driven decision-making. In the realm of networks, robots can serve as tangible entities to simulate and understand network configurations and protocols. Similarly, in embedded systems and artificial intelligence, the interactive nature of robots provides a dynamic platform for experimenting with algorithms, sensors, and actuators. This approach not only enhances the understanding of programming principles but also cultivates problem-solving skills, algorithmic thinking, and the ability to debug and optimize code in a tangible context. Moreover, the interactive nature of working with robots fosters a dynamic and engaging learning environment, inspiring creativity and collaboration among students. In what follows we highlight these specific benefits and demonstrate how the inclusion of robots contributes to a more comprehensive and effective educational experience in the field of computer programming.

### 4.1. Supported Courses

The design of a study curriculum is a complex process that requires several steps such as the definition of its objectives, the assessment of student needs, the review of

existing curricula, the actual development and testing of the curriculum, and the final implementation. This complex process is still beyond the scope of the current study, which however takes the courses that are usually found in computer science curricula in the last years of undergraduate programs or at the postgraduate level and proposes how they can be taught using a single educational robot. Microprocessors, sensors, and actuators play an important role in the development of solutions for real-world problems. Educational robots offer a valuable tool for enhancing computer science-related activities. In pursuit of this objective, we have chosen to employ FOSSBot and integrate its functionalities into selected courses. Since the software of the FOSSBot is built upon the Python programming language, we have established Python libraries to facilitate course activities. Furthermore, the hardware components of the robot are integral in shaping the nature of these activities. Subsequently, we delineate specific activities associated with the hardware components of the robot. The key steps of our methodology are summarized in Figure 4 and in what follows we explain how eleven core computer science courses can be supported in their activities during the classes using the robot.

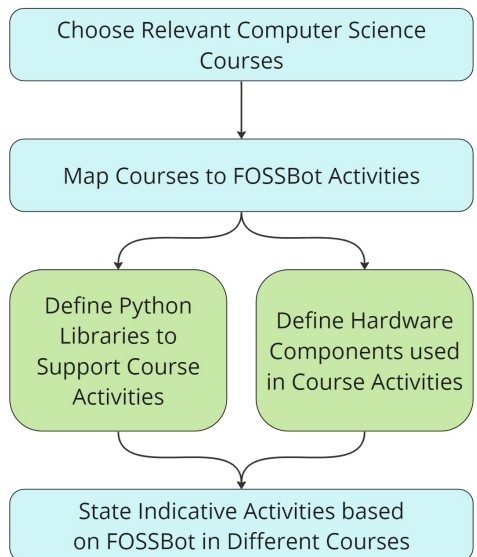

**Figure 4.** The Key Steps in the proposed methodology are outlined.

### 4.1.1. Electronics

Circuits and electronics form the foundation of electronic devices. These devices operate by allowing current to flow through wires, following the principles of Ohm's Law, which relates current, resistance, and voltage. Before programming a robot, a circuit is designed to connect various components such as sensors, actuators, controllers, and a battery using electronic elements like wires, capacitors, transistors, diodes, and resistors. Students learn to build hardware components, construct circuits on the robot, and set parts in motion. Students gain hands-on experience controlling current flow with resistors, connecting sensors and actuators to the microprocessor [31,32], and delve into basic programming and embedded systems design. Those new to programming can use predefined Blockly blocks to instruct FOSSBot parts, while those with programming skills can create new blocks for Raspberry Pi control. Through the course, students develop proficiency in designing and constructing circuits for robots, improve their skills in connecting sensors, and configuring actuators. They gain hands-on experience with fundamental electronic elements such as resistors, capacitors, transistors, and diodes, to optimize circuit performance. The skills acquired in this course are particularly useful in industries focusing on electronic device development, robotics, and embedded systems.

### 4.1.2. Embedded Systems

Embedded systems perform specific tasks in everyday objects, combining hardware and software with limited resources in memory and processing power. Microcontrollers, microprocessors, inputs, outputs, and memory work alongside programming languages, operating systems, and applications. These systems interact with the environment to control hardware through software. In educational contexts, microcontrollers and microprocessors are commonly used to teach embedded systems. Platforms such as Field-Programmable Gate Arrays (FPGA), Arduino, BeagleBone, and Raspberry Pi are popular choices, depending on project requirements. In [31,32], Arduino is employed for teaching embedded systems, while Raspberry Pi, the core of the FOSSBot, is a versatile tool for executing a wide range of tasks. The Raspberry Pi's General-Purpose Input/Output (GPIO) pins enable communication with sensors and actuators. Learners can also write software and applications directly on the Raspberry Pi, eliminating the need for an external computer. Activities for learning embedded systems with FOSSBot include interfacing with GPIO pins for data input/output and integrating hardware and software within the same device. In this course, students gain practical skills in developing systems that combine hardware and software to perform specific tasks with limited resources. They work with microcontrollers and microprocessors, learning to interface with inputs, outputs, and memory, while also exploring platforms like Arduino and Raspberry Pi. These activities equip them with the ability to design efficient systems that control hardware through software. Successful graduates can pursue roles as embedded systems engineers, IoT (Internet of Things) specialists, or firmware developers, where their proficiency in combining hardware and software in resource-constrained environments is crucial. Industries such as consumer electronics, automotive, and medical devices value these skills for designing smart and efficient systems.

### 4.1.3. Python Programming

Python is a multi-purpose programming language with applications in machine learning, data science, web development, and more. Its simplicity makes it a popular choice for teaching fundamental programming concepts, data structures, algorithms, and object-oriented programming in secondary and high education [34]. FOSSbot is designed around the Raspberry Pi microprocessor, which supports Python programming. Students can learn Python by interacting with physical sensors and issuing instructions based on the sensor data [8]. The robot's actuators offer full control, allowing learners to experiment visually and test various functionalities [35]. For instance, ultrasonic and infrared sensor data can be used to control motor wheels while teaching iterations, and light-dependent resistors (LDR) data can govern RGB LEDs using if/else conditions. The Python programming course equips students with versatile programming skills applicable to machine learning, data science, and web development, fostering hands-on learning with FOSSBot's Raspberry Pi support. Proficiency in Python opens diverse career opportunities in software development, data science, and artificial intelligence, providing graduates with valuable skills sought across industries.

### 4.1.4. Algorithms

An algorithm is a set of instructions used for efficiently performing a task and solving a problem in a specific order. In computer science, algorithms are typically implemented in software, which consists of code executed on processor-based hardware. However, they can also be realized in programmable hardware (e.g., FPGAs) or a combination of both. When building a robot, algorithms are applied to do a variety of tasks. Students are involved in the programming process of the robot so they could do some tasks like process data decision making. Algorithmic tasks in FOSSBot are written in the microprocessor and GPIO pins are used to work with input and output directly in hands-on activities. Many activities can be done by giving instructions. Graph algorithms can be given for finding better routes when the robot moves [25]. Through an Algorithm Design course, students gain

proficiency in crafting efficient sets of instructions for task execution and problem-solving, with a focus on practical application in robotics. They learn to implement algorithms in the microprocessor and GPIO pins of FOSSBot, honing skills in data processing, decision-making, and applying graph algorithms for optimized route planning. Proficiency in algorithm design opens doors to careers in software development, robotics, and artificial intelligence, where graduates can contribute to creating innovative solutions. Whether in tech companies, research institutions, or industries leveraging automation, these skills are highly valued for their broad applications, preparing students for roles as software engineers, algorithm developers, or robotics specialists.

### 4.1.5. Data Science (Data Mining)

Data science applies computational and statistical techniques to extract insights and knowledge from data, while data mining, a subfield of data science, focuses on uncovering patterns and relationships in data. Data from the physical environment varies in form and size. Algorithmic and statistical models help identify data trends, and relationships, and make predictions. Educational robots utilize sensors to collect environmental data, which is then analyzed and visualized. Machine learning models are developed and tested for predictive purposes. Another approach to using educational robots in data science involves collaborative learning, allowing students to collaborate on projects and share findings. Activities in data science courses encompass data acquisition, analysis, and visualization from sensors. Once the data is processed and analyzed, it can be used to construct and execute models [36] that control actuators by communicating through GPIO pins. The course equips students with computational and statistical skills to extract insights from diverse data forms, utilizing algorithmic and statistical models to identify patterns and relationships. Students engage in hands-on activities involving sensor data acquisition, analysis, and visualization, culminating in the construction and execution of models controlling actuators through GPIO pins. Proficiency in data science, especially in data mining, opens avenues to careers in industries reliant on data-driven decision-making, such as finance, healthcare, and technology. Graduates can pursue roles as data scientists, analysts, or machine learning engineers, where their expertise in uncovering patterns and making predictions from diverse data sets is invaluable for optimizing processes and driving innovation.

### 4.1.6. Artificial Intelligence

Devices and systems may get smarter by applying intelligence. Artificial intelligence uses previous data to make better decisions such as prediction, recommendation, and learning from experience. Models are trained to perform a variety of tasks, which would be applied in the machines to think and act like a human. A robot is a good example of where artificial intelligence could be applied. Data collected by the sensors are trained and used to make decisions which will be applied to actuators. Robots can collaborate to do some tasks together by sharing data they measure. The Raspberry Pi microprocessor is an excellent tool that allows to running of lightweight artificial intelligence and machine learning models using specific Python libraries. To provide a few examples, reinforcement learning models can be applied to find the best route among obstacles, and supervised learning models can be used to help the robot make decisions based on sensor input, e.g., to detect and avoid obstacles, to follow a visual or thermal signal, etc. [28,29,37]. In the Artificial Intelligence and Machine Learning course, students acquire the ability to enhance devices and systems by leveraging previous data to make informed decisions, applying models trained for tasks such as prediction and recommendation. They gain hands-on experience with Raspberry Pi, using Python libraries to implement reinforcement learning for optimal route finding and supervised learning for tasks like obstacle detection and avoidance in robots. Proficiency in AI and ML opens pathways to careers in industries seeking innovative solutions, such as robotics, automation, and technology. Graduates can pursue roles as AI engineers, machine

learning specialists, or robotics developers, where their skills in creating intelligent systems and enabling machines to learn from data contribute to advancements in various sectors.

4.1.7. Natural Language Processing

Natural Language Processing (NLP) is a field of Artificial Intelligence that focuses on the interaction between humans and computers. A natural language is a key input that must be understood by computers and converted into meaningful and useful text. Common tasks and techniques using NLP in educational robots are speech recognition and machine translation. Microphone or sound sensors attached to the robot measure human speech and are interpreted into text [36]. Robots can generate text and give information by interacting with learners. Instructions from learners will be converted into a code to do some activities. A sound sensor is available on FOSSBot and its data can be fed to NLP models running on the Raspberry Pi microprocessor. In this course, students develop skills in enabling meaningful communication between humans and computers, with a focus on converting human speech into text. They engage in hands-on activities using educational robots to implement NLP techniques like speech recognition and machine translation, fostering the robot's ability to understand and generate text through interaction with learners. Proficiency in NLP opens career paths in industries where human-computer interaction is crucial, such as virtual assistants, customer support, and language processing applications. Graduates can pursue roles as NLP engineers, conversational AI developers, or language technology specialists, contributing to the development of intelligent systems capable of understanding and generating human language.

4.1.8. Computer Vision

Robots also interact with their environment by capturing and analyzing images and videos to understand and interpret visual information. Computer vision is a field of study that focuses on this information. Common tasks of computer vision are object recognition, tracking moving objects, and understanding image content. Data for these tasks are collected by cameras available on the robot [36]. The operating system installed on the microprocessor of the FOSSBot allows it to run and execute computer vision algorithms and techniques for a variety of tasks. The robot can recognize humans or objects and can interact with them. The other activity is object following and driving by tracking it [10]. Learners also can program the robot so that it recognizes objects, road signs, and traffic lights when driving through the road. The Computer Vision course empowers students to enable robots to comprehend and interpret their environment through image and video analysis. Focusing on tasks like object recognition and tracking, learners utilize cameras on the robot to collect data. They gain practical experience in implementing computer vision algorithms, enabling the robot to recognize and interact with humans or objects, track moving entities, and perform activities like object following and driving with road signs and traffic light recognition. This experience is valuable in industries where visual information interpretation is pivotal, including autonomous vehicles, surveillance, and image processing. Graduates can pursue roles as computer vision engineers, image processing specialists, or robotics developers, contributing to advancements in fields requiring intelligent visual perception systems.

Running computer vision tasks on an educational robot would contribute valuable insights into managing multimedia data in the edge, and considering energy efficiency and load balancing which are critical factors in optimizing resource utilization in such scenarios. Students can explore the technical aspects of energy-efficient resource management and load balancing in multimedia-rich robotics and will get a solid foundation for understanding how to optimize resource usage, enhance performance, and ensure the efficient operation of educational robots with multimedia capabilities [38].

### 4.1.9. Cloud Computing

Cloud computing is used for accessing diverse computing services such as storage, servers, analytics, networks, and software over the internet making the connection faster and more reliable. Principles of cloud computing include designing, building, and managing computing systems. Also, it allows direct connection to the services and applications without active management of the user. The educational robots could be connected to the cloud directly and share information with it. The WiFi FOSSBot connection allows one to control it remotely, but can also be used to connect to the cloud and use remote resources to carry out computationally demanding tasks, data storage, or data analysis. Similarly, the robot can be controlled remotely over the cloud using cloud APIs and allow cloud-based robotic applications to be built and deployed. Although not yet supported, several FOSSBots can communicate and work together to further enhance their performance and efficiency. The Cloud Computing course equips students with the skills to leverage cloud services, enabling educational robots to access computing resources like storage, servers, and analytics over the internet. Students learn principles of cloud computing, including designing and managing computing systems, and facilitating direct connections to services without active user management. They explore the potential of connecting robots to the cloud for remote control, computational tasks, data storage, and analysis, enhancing their understanding of cloud-based robotic applications. Proficiency in cloud computing is highly valuable across industries, particularly in technology and data-driven sectors. Graduates can pursue roles as cloud engineers, system architects, or data analysts, where their ability to integrate educational robots with cloud resources contributes to efficient, scalable, and remotely controlled robotic systems.

### 4.1.10. Cybersecurity

Security risks must always be considered in every aspect of personal and professional lives that computers, the internet, and other digital technologies rely on. Digital devices, networks, and data are protected from digital attacks and damage. Most of the attacks come through the network when they are connected to the Internet. The cybersecurity course teaches principles of cybersecurity including information security, firewall implementation, and detection of intrusions. Unauthorized access to the system and information must be prevented against vulnerabilities, malicious programs, destruction, and data theft. The educational robot is accessible through the network so that users can remotely interact with it. Students will be able to practice protecting the robot programming stack and the microprocessor from unauthorized access, creating and enforcing password security rules, installing firewalls, etc. These are some of the activities that can be used for teaching cybersecurity using FOSSBot. In the Cybersecurity course, students learn to protect digital devices and data, focusing on principles like information security and intrusion detection. They practice securing the robot's programming stack and microprocessor against unauthorized access, enforcing password rules, and implementing firewalls. Proficiency in cybersecurity is vital in various industries for securing digital systems and sensitive information. Graduates can pursue roles as cybersecurity analysts, network security specialists, or information security officers, contributing to the growing demand for professionals adept at safeguarding interconnected technologies.

### 4.1.11. Internet of Things

Building IoT applications a broad requires knowledge of engineering and computer science. IoT consists of components, a network, and the cloud, much like FOSSBot's architecture. They communicate similarly. Hardware includes a microcontroller, sensors, and actuators. Sensor data from the physical environment is transmitted to the cloud via a network. The computed data is sent back to the device to take action using actuators. FOSSBot provides various sensors, actuators, and wireless technologies. For example, users can measure wheel distance and odometers [25,31]. Data is transmitted via Wi-Fi or Bluetooth available in robot microprocessors. Analog and digital data on sensors and

actuators use communication protocols within the robot. Edge computing also occurs in the microprocessor. Educational robots in IoT courses enable collaborative group work and project-based learning, empowering students to create customized robots based on FOSSBot's electronics [39–41]. The IoT course teaches students to create applications mirroring FOSSBot's architecture, emphasizing hardware, sensors, and cloud communication. Using the robot's wireless tech, students practice real-world applications like measuring wheel distance. Proficiency in IoT is valuable for roles in engineering and computer science, enabling graduates to work on innovative projects. The course, using educational robots, fosters collaborative learning and project-based skills, preparing students for careers as IoT developers or hardware engineers.

Another important aspect of IoT that can be examined using the robot is load balancing and QoS. Understanding how to manage resources efficiently and ensure quality service delivery is essential for optimizing the performance of IoT systems especially when dealing with a network of interconnected devices like educational robots. Students can delve into the technical aspects of implementing load balancing and QoS mechanisms in IoT settings and get a deeper understanding of how these concepts can be applied to improve a robot's functionality, network performance, and overall user experience [42].

*4.2. Specific Libraries That Extend FOSSBot in Supporting Course Activities*

Software libraries are collections of code that solve specific problems or support certain tasks such as security, networking, data management, etc. They can extend the robot's capabilities and make it useful for different purposes, they can simplify work in user processes for the users who are not very experienced in programming. The software stack of FOSSBot allows for loading a variety of libraries and supporting different course activities. FOSSBot libraries run on the Raspberry Pi microprocessor and have to be implemented in Python. For example, the RPi.GPIO library is written in Python and allows working with the GPIO pins in order to get input values from sensors and send the output results to the actuators. The library can be used to provide hands-on experience in programming, electronics, and embedded systems courses. Gpiozero, gpiopi, pigpio, and WiringPi are additional python libraries that allow the communication of the microprocessors with the sensors and actuators using the SPI and I2C communication protocols. These libraries can support activities in embedded systems and IoT courses. The sci-kit learn, matplotlib, and seaborn libraries are widely used in data science and machine learning course activities and can be used in FOSSBot to allow working with the data collected by the sensors. They can be used in combination with numpy, scipy and pandas libraries to process data and allow students to implement and test algorithms that handle sensor data and apply data science principles in practice. Several code libraries can be employed for processing the input from the camera (e.g., OpenCV) or the microphone and allow students to practice human-robot interaction tasks or to implement smart solutions that test AI algorithms. Libraries such as TensorFlow Lite and PyTorch or cloud-based solutions such as Edge impulse, Google Assistant API, ThingSpeak can also be used in the same direction. More demanding tasks can be ported on the cloud allowing students to practice the use of AWS, Azure, or other cloud resources. Natural Language Processing (NLP) libraries, such as NLTK or spacy, can be used to develop text interfacing solutions with the robot, helping students to learn NLP concepts and solutions in practice. Finally, basic cybersecurity principles can be tested in practice on the open robot using OS distributions (e.g., Kali Linux) that are targeted to penetration testing and forensics. The list of courses and the respective Python libraries that can be used in FOSSBot are depicted in Table 2.

**Table 2.** The curriculum courses and the Python libraries that can be employed in activities.

| Course | Library | Sensors and Actuators | Activity |
|---|---|---|---|
| Electronics | Gpiozero, gpiopi, RPi.GPIO, WiringPi, pigpio | all sensors and actuators | Wire microprocessor with sensors and actuators using transistors, capacitors, resistors, and diodes, program the microcontroller |
| Embedded Systems | Gpiozero, gpiopi, RPi.GPIO, WiringPi, pigpio | all sensors and actuators | Raspberry Pi can be used to teach embedded systems by orchestrating the operation of sensors and actuators via the GPIO pins |
| Python programming | Gpiozero, gpiopi, RPi.GPIO, WiringPi, pigpio | all sensors and actuators | Teach basic programming, functions that perform specific tasks, extend the core FOSSBot class |
| Algorithms | Numpy, scipy, pandas | sensors, motor wheels | Give instructions to control the robot, implement navigation (search space) algorithms based on graph theory (for finding shortest path) |
| Data Science | scikit, matplotlib, seaborn | sensors for collecting data | Collect data by sensors, learn data acquisition techniques, visualize data in plots |
| Artificial Intelligence | TensorFlow, Edge Impulse, OpenCV, Google Assistant API | microphone, camera | Use the camera and the microphone to interact with a learner, perform object detection and avoidance |
| Natural Language Processing | Google Assistant API, NLTK, spacy | microphone, speaker | Convert voice instructions text and code. Interact with the user with voice. |
| Cloud Computing | AWS, Azure, ThingSpeak, Edge Impulse | all sensors and actuators | Control the robot remotely, run complex processing tasks in the cloud, run data transmission and storage tasks |
| Computer Vision | Open CV, TensorFlow, PyTorch | camera | Object detection, object following, inference using pre-trained models, balance execution between the edge (FOSSBot) and the cloud. |
| Cybersecurity | Kali Linux OS | wi-fi, bluetooth | Protect the robot against vulnerabilities, unauthorized access |
| Internet of Things | PID, gpiopi, RPi.GPIO, WiringPi, pigpio, Gpiozero | all sensors and actuators | System control and processing, explain IoT architecture, communication protocols, data transmission through a network |

*4.3. Indicative Activities*

A variety of sensors and actuators enables the robot to be used in different activities. The available sensors are gyroscope, accelerometer, odometer, photoresistor, infrared sensor, and ultrasonic sensor. The available actuators are the DC motor, the speaker, and the LED. Since FOSSBot is an open design it can also be extended with other sensors and actuators based on the purpose of use in activities. A camera, microphone, and LIDAR sensors are a few examples that can be used with FOSSBot. Single sensors can be used in several types of applications. For example, an odometer can be involved in line following activity as well as for measuring the traveled distance. In Table 3, we provide a list of FOSSBot's hardware components and describe activities that can be based on each one of them.

**Table 3.** Activities associated with hardware components of the robot.

| Hardware Components | Activities |
| --- | --- |
| Raspberry Pi | Embedded systems programming, input and output data via GPIO pins |
| Gyroscope | Measure orientation of the robot, provide navigation, maintain reference direction |
| Accelerometer | Measure changes in speed |
| Odometer | Line following, measure distance traveled |
| Photoresistor (LDR) | Measure light level, control LED using light level |
| Infrared sensor | Detect motion, measure distance from obstacle, line following |
| Ultrasonic sensor | Measure distance between an object and the robot, object following, obstacle avoiding |
| Camera | Object detection, object following, obstacle avoiding, physical security of the robot |
| Microphone | Voice detection, voice instructions |
| LIDAR | Provide an accurate representation of the surveyed environment, object following |
| DC motor | Implement movement instructions, navigate using wheels, control acceleration and robot turning |
| RGB LED | Use it as a visual alarm, as a traffic light, learn how to change the output using basic programming |
| Speaker | Communicate with a learner |
| Wi-Fi | Data transmission, remote control |
| Bluetooth | Data transmission over short distances, remote control |
| IR remote | Communicate over a short distance for remote control |

The teaching of courses in the curriculum can be enriched with hands-on activities that exemplify the practical implementation of each course. In the Electronics course, students wire the Raspberry Pi microprocessor with sensors and actuators, gaining practical experience in working with electronic components like transistors, capacitors, resistors, and diodes. They further enhance their skills by programming the microcontroller to execute specific tasks. In the Embedded Systems course, students orchestrate the operation of sensors and actuators using the GPIO pins on the Raspberry Pi, providing a tangible understanding of embedded systems concepts.

Moving to the Python Programming course, students can learn basic programming skills by utilizing the core robot class and writing functions for specific tasks. The Algorithms course focuses on implementing navigation algorithms based on graph theory, showcasing how students provide instructions to control the robot's movements and find the shortest path. In the Data Science course, learners use sensors to collect data and apply data acquisition techniques, gaining hands-on experience in visualizing data through plots. This practical application reinforces their understanding of data science concepts.

In the Artificial Intelligence course, the integration of a microphone and camera allows students to interact with the robot using voice commands and perform object detection and avoidance. This real-world application demonstrates how AI models can be applied to enhance the robot's capabilities and interaction with users. The curriculum further extends to all other courses integrating specific libraries and activities tailored to the respective domain for a comprehensive and hands-on learning experience. This emphasis on practical examples across various courses ensures that students develop tangible skills applicable to real-world scenarios.

## 5. Evaluation

The evaluation of an educational intervention typically employs qualitative and quantitative criteria to provide a comprehensive understanding of its effectiveness. Qualitative criteria assess aspects such as participant satisfaction, perceived impact, and the richness of the learning experience through methods like interviews, observations, and open-ended

surveys. On the other hand, quantitative criteria involve measurable outcomes such as academic performance, retention rates, and standardized test scores, offering numerical data to quantify the intervention's success and its impact on learners' knowledge and skills. Integrating both types of criteria allows for a holistic assessment, capturing both the subjective and objective dimensions of the intervention's effectiveness.

The quantitative evaluation requires the intervention to be applied in a formal educational framework, using control and study groups of students and comparing their performance before and after the application of the new practice. Respectively, the qualitative evaluation can be applied in a less strict format, with interviews or surveys of the students and professors who employed the robot in all possible educational activities. Since the proposed approach is still at an early stage, we decided to evaluate the robot both in a formal and a less formal educational framework. First, we used the educational robot in terms of an academic course on IoT. Secondly, we conducted several students who have been engaged in various activities with FOSSBot, which cover a wide range of concepts, from electronics and IoT (e.g., in FOSSbot design and assembly) to programming (e.g., of the FOSSbot library and UI) and machine learning (e.g., by using FOSSbot to learn reinforcement learning concepts and algorithms).

*5.1. Evaluation in Terms of an Academic Course*

To evaluate the effectiveness of the proposed educational robot, students were divided into two groups: an experimental group comprising 20 students, and a control group of 13 students. All students attended the same lectures but participated in different practice sessions, which were based on three activities that differed between the groups and provided a comprehensive exploration of IoT concepts and applications. The control group persisted with the traditional method and utilized the Packet Tracer desktop application to comprehend the operational principles of the IoT ecosystem. In contrast, the experimental group embarked on activities associated with the proposed robot and encompassed tasks such as remotely controlling the robot via Bluetooth, constructing a line follower robot utilizing IR sensors, and managing the robot to transmit sensor data to the cloud via wireless communication.

The tasks and hardware components incorporated in the activities were consistent for all students within a given group. Students were asked to create a circuit, establish connections between the hardware components, and program the device (or the robot, in the case of the experimental group). They were permitted to use any library for this purpose. Their work was evaluated based on several criteria, including the quality of the code they wrote, the circuit they created, and the performance of the resulting setup (device or robot). This comprehensive evaluation approach ensured a fair and thorough assessment of each student's understanding and application of the concepts taught in the course. The final average grade for the control group was higher (59/100 points) than that of the experimental group (46.1/100 points).

Apart from their performance during the course, a project was assigned to both groups at the end of the course to assess students' comprehension of the course topics and activities. The project requirements were identical for both groups. Students were free to choose any topic but were asked to solve a real-world problem that aligns with the broader IoT ecosystem, utilizing microcontrollers, sensors, and actuators. The proposed solution was expected to encompass the construction of a device, the establishment of a network using communication technology, and the development of an application capable of monitoring or controlling the device. The evaluation of the solutions was based on the identification of the problem, the complexity of the solution, and the integration of each component of the solution within the IoT architecture. This comprehensive evaluation approach ensured a thorough assessment of each student's understanding and application of the concepts taught in the course.

Most of the students from the experimental group choose to continue working with the robot, undertaking various tasks. Some students developed an Android application to

remotely control the robot, while others utilized different cloud applications to measure data for environmental monitoring using the robot. Despite these lower scores in course activities, the experimental group surpassed the control group in the course project, scoring 60.3/100 against 53.4/100 points on average. The results of the two evaluation stages (i.e., during the educational activities and in the final project) show that the students in the experimental group significantly improved their performance, whereas the performance in the control group dropped. This could be an indication that the use of the educational robot can boost the creativity of the students and at the end of the process can help them achieve higher performance than using traditional tools.

*5.2. Evaluation of Experiences with FOSSBot*

To qualitatively evaluate the opinions of various students who used the educational robot in various contexts, we contacted former students at SDU and Harokopio University of Athens and asked them to fill in a survey that captured their experiences in working with the robot. Fourteen (14) students in total responded to the questionnaire, and the main findings are summarized in the following:

- The students have a different engagement span with the robot, from less than a month (14.3%) to a year or more (57.1%).
- Most of the students the robot as undergraduate students (85.7%), and a few used them as part of their PhD research (14.3%).
- More than half of the students used the robot within a course (57.1%), but there are still uses in the framework of degree projects (21.4%), or in other activities such as contests or summer camps.
- The students have practiced mainly IoT, Programming, and Electronics subjects, whereas some of them used the robot in simulation projects.
- Among others, they foresee the use of FOSSbot in more courses including robot design and 3D-printing, machine learning and AI, and STEM education.
- All of the students found that the robot increased a lot their engagement in the subject, and most of them (93%) found it very enjoyable.
- Last, but more importantly, almost all of them (93%) believe that the robot helped them a lot to get more knowledge in the respective subject they practiced.

Among the free-text suggestions for extending the use of FOSSbot in other courses and in more activities, the students mentioned the need for computer vision and machine learning capabilities of the robot that will help them practice the respective subjects and develop solutions in real-world tasks such as obstacle avoidance.

When reflecting on the broader pedagogical impact that the introduction of educational robotics can have on tertiary education, and more specifically into teaching computer science concepts, it becomes evident that the proposed approach is not only able to enhance traditional learning methods but also seeks to cultivate essential skills and attributes crucial for success in the modern workforce. One significant pedagogical impact of our approach is the promotion of active and experiential learning. Through hands-on activities with the educational robot, students are actively engaged in constructing knowledge, problem-solving, and applying theoretical concepts in practical scenarios. This active engagement allows a deeper understanding of computer science principles and promotes critical thinking and creativity, which are essential skills in today's rapidly evolving technological landscape. Furthermore, the use of educational robotics encourages interdisciplinary learning and collaboration. By integrating concepts from various domains such as electronics, programming, and IoT, students gain a holistic understanding of complex systems and are better equipped to tackle real-world challenges that require interdisciplinary approaches. Moreover, collaborative tasks, such as designing and programming the robot, promote teamwork and communication skills, which are essential in professional settings.

Another important pedagogical aspect is the promotion of innovation and entrepreneurship. By providing students with opportunities to explore and experiment with emerging technologies like robotics, our approach cultivates an entrepreneurial mindset and em-

powers students to innovate and create solutions to real-world problems. This not only prepares them for careers in technology but also equips them with the skills and mindset necessary to drive positive change and contribute to society. Overall, this integration has the potential to revolutionize computer science education by fostering active learning, interdisciplinary collaboration, and innovation. By embracing this approach, educators can better prepare students for the challenges and opportunities of the digital age, equipping them with the skills, knowledge, and mindset needed to thrive in a rapidly evolving technological landscape.

## 6. Conclusions and Future Directions

This work introduced a holistic approach to the use of educational robots in teaching Computer Science courses. With the main objective being to demonstrate the feasibility of this approach, we explained how an open-source robot could be the main platform for developing activities for a wide range of courses. An open-source and open-design educational robot with multiple sensors and actuators allows for supporting different courses, with extensions and customization when necessary. A one-year computer science curriculum, containing courses that cover electronics, networks, artificial intelligence, data management, and programming, is feasible and can be delivered at the post-graduate level using FOSSbot as the main platform for all the experimental educational activities. Building on the open nature of the robot, we propose specific Python libraries that extend FOSSBot in supporting the course activities and explain how the various robot components (the microprocessor and the sensors and actuators) can be used in support of the activities.

The results of our work demonstrate that the gap in the literature of educational robotics, which is related to the fragmented solutions developed on a per-course basis, can be bridged using an open-source robot. Adapting the robot to the nature and needs of each course, taking advantage of its sensors, and designing and developing the appropriate activities are important steps for the implementation of the proposed approach.

In this work, we evaluated the performance of our intervention both in a university course and in an informal educational context and compared the students' performance with or without using the educational robot for practice. The quantitative evaluation in terms of a course can show whether the use of the robot in educational activities improved the performance of students. In addition, utilizing a small sample of students who engaged with the intervention in an informal and extra-curricular manner can offer valuable insights and perspectives that might not be captured through formal participation. The students represented a diverse range of backgrounds, learning styles, and interests, providing a nuanced understanding of the intervention's impact beyond structured educational settings. Moreover, their voluntary engagement signals a heightened level of intrinsic motivation and interest, potentially reflecting a deeper level of investment in the intervention's outcomes. With a composite evaluation that combines the merits of quantitative metrics (e.g., performance) and qualitative criteria (e.g., engagement), we gained a more holistic understanding of the intervention's effectiveness, capturing the feedback of those who have actively engaged with it and we are now ready to evaluate the proposed approach in a wider range of courses and topics.

Following the design of the courses that comprise the curriculum, the next steps of our work will focus on extending the FOSSBot software stack with software implementations that use sensors as described in course activities and modify the design to include more sensors such as a microphone, a LIDAR, and a camera that make it more powerful compared to other educational robots. We are also working on providing support to the popular operation system for robots ROS (Robotic Operating System) which is supported by the Raspberry Pi microprocessor and will add to the expandability of the robot. ROS contains libraries that standardize the use of robot's sensors and actuators and make the communication between software and hardware components simpler and easier. In the absence of a physical robot, virtual simulations through ROS provide an alternative avenue for students to explore and experiment with various aspects of robotics, fostering a deeper

understanding of algorithms and software implementation. Last but not least, we aim to collaborate with educators in various universities, who specialize in the education of each course using robotics to develop specific activities and educational material and include FOSSbot in their educational practice.

With this work we hope to empower the use of open-source and open-design robots in tertiary education, for the development of holistic approaches in the use of robots for supporting multiple courses and if possible whole curricula, thus fostering an inclusive and innovative educational landscape that prepares students for the evolving demands of the digital era.

**Author Contributions:** C.C. did the design of the robot and supported several students in using it for their diploma thesis or extra-curriculum activities, with the supervision of I.V., Z.M. assembled the robot and used it in the educational process and the evaluation with the supervision of M.Z., Z.M. and Y.H. drafted the manuscript, and I.V. with M.Z. prepared the final version. All authors contributed to the description of courses and activities that can be supported by the robot. All authors have read and agreed to the published version of the manuscript.

**Funding:** This work has been partially supported by the Erasmus+ (KA220-HED-Cooperation partnerships in higher education) project S.T.E.P.S. under grant agreement No. 000165711.

**Data Availability Statement:** Dataset available on request from the authors.

**Acknowledgments:** The authors would like to thank the Greek Free Open Source Software community (GFOSS) for supporting the FOSSBot project.

**Conflicts of Interest:** The authors declare no conflict of interest.

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
