# Peer review of "A Holistic Approach to Use Educational Robots for Supporting Computer Science Courses"

_computers, doi:10.3390/computers13040102_

Round 1

Reviewer 1 Report

Comments and Suggestions for Authors

The title of the paper  "A Holistic Approach for the Use of Educational Robots in the Development of IT Curricula" is misleading as nowhere in the paper do you consider "the development of curricula".  IT may also not be quite what you are referring to -- the modules or courses you mention are possibly more Computer Science than IT?   

You actually say so yourself in Section 4 where you say "The design of a study curriculum is a complex process....This complex process (curriculum design)  is still beyond the scope of the current study, which however takes the courses that are usually found in computer science curricula in the last years of undergraduate programs or at the postgraduate level and proposes how they can be taught using a single educational robot.

The following title might thus be more appropriate for your paper: "Exploring the integration of educational robots in higher education".

In Section 4 (line230) you say:"Educational robots offer a valuable tool for enhancing computer science-related activities. In pursuit of this objective, we have chosen to employ FOSSBot and integrate its functionalities into selected courses." You need to specify the objectives of your study in the introduction or have a research question that you addressThis is the only time in the paper that you used the word "objective" .  You also need to motivate why you chose to discuss those specific computer science courses. 

In the conclusion you need to revisit the objectives of the study. You will also have to rewrite your abstract, it is not concise enough.

The following format/template is suggested for a Scientific Article---try to align your article to match it:

Abstract

State the principle objectives and scope of the investigation;

Describe the methodology employed;

Summarize the results;

State the principle conclusions reached. It should not give any information or conclusion that is not stated in the paper.

Introduction

A good introduction will contain the following.

(1) It should present a clear outline of the nature and scope of the problem investigated;

(2) It should state the hypothesis being tested; the objectives or research question

(3) It should give the motivation for doing the research and the background knowledge that is considered essential for the reader to understand the paper;

(4) It might explain why the problem was studied;

(5) It should briefly review the pertinent literature;

(6) It should briefly state the method of investigation;

(7) It should give the principal results of the investigation.

Materials & Methods

In this section the author should give enough detail that a competent worker can repeat the experiments described. The careful writing of this section is critically important because scientific convention requires you’re your results be reproducible- thus you must provide the means for another researcher to reproduce your experiments. A reviewer of your paper will read this section with close attention, especially if your results controversial.

If your method is new, you must give all the needed details. If the method has been published previously in a primary journal, you need give only the literature reference. If you modified a familiar procedure, describe what you did that was new.

State whatever statistical procedures you used on your results.

Results

There are two kinds of components of the results section. First, you should present the data. Second, you should give some overall description of the experiments (providing the ‘big picture’).

Since the entire paper of the results they must be presented with great clarity, succinctness and, as far as possible, simplicity. Decide on the most efficient and logical way of showing your findings, and avoid duplication. Do not describe in detail in the text what is obvious from a table or figures tell the main story, the text need mention only the highlights.

The results section may well be relatively short, especially if it is sandwiched between a well-written Materials and Methods section and followed by a well-written Discussion.

Each figure and table should have a succinct, self-contained and accurate caption.

It is sometimes sensible (and honest!) to state what you did not find under the conditions of your experiments.

Conclusion

This is often the hardest section to write. Many are too long and verbose. Apart from the abstract, it is the section that most influences an editor or reviewer when first examining a paper. Remember, in a good Discussion, you discuss, and do not recapitulate, the results.

The essential features of a good Discussion are:

1.    Try to present the principles, relationships and generalizations shown by the results.

2.    You should state whether the data support the hypothesis being tested

3.    You point out any exceptions or any lack of correlation among your results, and define unsettled matters.

4.    Show how your results and interpretations agree (or contrast with) previously published work

5.    Discuss the theoretical implications of your work, as well as any possible practical applications

6.    State your conclusions (and possibly summarize your evidence for each conclusion).

7.    It is most important to indicate the significance of the work reported. A reader must not be left asking, “So what?”

Comments on the Quality of English Language

A few minor comments were made on the document.

Author Response

We would like to thank the reviewer for the detailed comments and suggestions, as well as for the comments directly on the PDF file. We incorporated everything in this revised manuscript, using blue font to indicate the changes.

Comment 1: The title of the paper  "A Holistic Approach for the Use of Educational Robots in the Development of IT Curricula" is misleading as nowhere in the paper do you consider "the development of curricula".  IT may also not be quite what you are referring to -- the modules or courses you mention are possibly more Computer Science than IT

You actually say so yourself in Section 4 where you say "The design of a study curriculum is a complex process....This complex process (curriculum design)  is still beyond the scope of the current study, which however takes the courses that are usually found in computer science curricula in the last years of undergraduate programs or at the postgraduate level and proposes how they can be taught using a single educational robot.

The following title might thus be more appropriate for your paper: "Exploring the integration of educational robots in higher education".   

Response 1: Thank you for this comment. We agree that the development of a curriculum is a more complex process, which probably does not fit easily in 15 pages. We would also like to focus on computer science and not refer to higher education in general. Since we are not exploring and experimenting with multiple robots that are available in the market but we build on our own open-source robot, we believe that exploring does not fit well in the title. So following your suggestions we revised the title to refer to computer science courses instead of IT curricula and used the term “support” instead of “development”. The new title is “A Holistic Approach to use Educational Robots for Supporting Computer Science Courses”.

Comment 2: In Section 4 (line230) you say:"Educational robots offer a valuable tool for enhancing computer science-related activities. In pursuit of this objective, we have chosen to employ FOSSBot and integrate its functionalities into selected courses." You need to specify the objectives of your study in the introduction or have a research question that you addressThis is the only time in the paper that you used the word "objective" .  You also need to motivate why you chose to discuss those specific computer science courses. 

Response 2: Thank you for this constructive comment. We updated the introduction to reflect the main objective of our work. We also added 3 references to papers that highlight the need for our work. The paragraph that we added is as follows:

The existing literature on educational robotics in tertiary education mostly focuses on using commercial robots or their virtual simulations for supporting specific courses and teaching specific subjects each time [39 ,40]. They usually lack a holistic approach in using the same robot to support all the courses of a curriculum and provide activities that fit the needs of each specific course. The main objective of the current work is to fill this gap by using an open-source robot and providing an approach for using it in multiple courses of a Computer Science curriculum. We believe that an open-source robot allows for more customization and expansions and can be the common basis for a holistic approach in the use of Educational Robots for supporting multiple courses of a curriculum. This provides better integration of emerging technologies -especially robotics- into higher education through actual practice and engagement [41].

The references we added follow:

“39. Spolaôr, N.; Benitti, F. B. V. Robotics applications grounded in learning theories on tertiary education: A systematic review. In Computers and Education, 2017, 112, 97-107.

  1. Atman Uslu, N.; Yavuz, G. Ö.; Kocak Usluel, Y. A systematic review study on educational robotics and robots. Interactive Learning Environments, 2023, 31(9), 5874-5898.
  2. Leoste, J.,; Jõgi, L.; Õun, T.; Pastor, L.; San Martín López, J.; Grauberg, I. Perceptions about the future of integrating emerging technologies into higher education—the case of robotics with artificial intelligence. Computers, 2021, 10(9), 110.”

Comment 3: In the conclusion you need to revisit the objectives of the study. You will also have to rewrite your abstract, it is not concise enough.

Response 3: In an attempt to match the updates performed in the whole paper, we followed your advice and rewrote the abstract to make it more concise and in accordance with the objective. We also revisited the main objective in the conclusions section and expanded the section accordingly.

Comment 4: The following format/template is suggested for a Scientific Article---try to align your article to match it:

Response 4: We would like to thank you for your extensive support in formatting our manuscript. We kept the main structure of the content, which we believe aligns with what you suggested. More specifically, the main material that we employ is the open source robot, which is the basis of our approach. The results of our work are in essence the descriptions of the courses that will be supported by the open robot, the specific libraries that will allow to expand the robot functionality and the list of indicative activities that can be supported. In this perspective, the results section is quite extended but definitely does not contain any numbers, since we didn’t run any experiments. However, it contains tables that summarize the components and activities supported. We changed the section titles and enhanced the content to match your recommendations. Since the changes are many and dispersed in the document we avoid listing them here. 

Reviewer 2 Report

Comments and Suggestions for Authors

Section 5 is missing many of the libraries you mention in Table 2, such as the OpenCV library, etc.

I would suggest being more explicit about why there is a need for robots in such a curriculum. What can be done better using robots? Okay in the field of electronics but why in the field of computer programming? I agree with your point, but make it clearer, in the article.

In line 280 you state "In computer science, algorithms can be implemented in both hardware and software or a combination of both". What do you mean by implementing the algorithm in hardware? All code is executed in hardware, I think that's not what you mean.

 The implementation of ROS is very critical. In case there is no real robot, all courses can be run, except for the electronics and embedded systems courses.

Author Response

We would like to thank the reviewer for the detailed comments and suggestions. We incorporated everything in this revised manuscript, using blue font to indicate the changes.

Comment 1: Section 5 is missing many of the libraries you mention in Table 2, such as the OpenCV library, etc.

Response 1: Thank you for noticing this. Following another reviewer’s comment we changed the numbering of our sections so the old section 5 is now subsection 4.2. As you suggested we added all the libraries listed in Table 2 to this subsection.

Comment 2: I would suggest being more explicit about why there is a need for robots in such a curriculum. What can be done better using robots? Okay in the field of electronics but why in the field of computer programming? I agree with your point, but make it clearer, in the article.

Response 2: Thank you for this comment, which gives us the opportunity to highligh the need for robots in teaching more courses apart from electronics. We added an introductory paragraph in section 4 that better justifies this need and explains what can be done better using robots.

“The main result of the proposed approach relies upon the unique advantages that robots bring to the educational landscape, not only in electronics but also across diverse domains such as programming, data science, networks, embedded systems, and artificial intelligence. By integrating robots into programming courses, students gain hands-on experience in translating code into real-world applications, reinforcing theoretical concepts through practical implementation. For instance, in data science, students can leverage robots to collect and analyze data, gaining insights into the practical applications of data-driven decision-making. In the realm of networks, robots can serve as tangible entities to simulate and understand network configurations and protocols. Similarly, in embedded systems and artificial intelligence, the interactive nature of robots provides a dynamic platform for experimenting with algorithms, sensors, and actuators. This approach not only enhances the understanding of programming principles but also cultivates problem-solving skills, algorithmic thinking, and the ability to debug and optimize code in a tangible context. Moreover, the interactive nature of working with robots fosters a dynamic and engaging learning environment, inspiring creativity and collaboration among students. In what follows we highlight these specific benefits and demonstrate how the inclusion of robots contributes to a more comprehensive and effective educational experience in the field of computer programming.”

Comment 3: In line 280 you state "In computer science, algorithms can be implemented in both hardware and software or a combination of both". What do you mean by implementing the algorithm in hardware? All code is executed in hardware, I think that's not what you mean.

Response 3: Thank you for this comment. Implementing an algorithm in software means that it is running in a processor-based hardware. In the hardware implementation of an algorithm (e.g. in programmable hardware or FPGA) a digital circuit is designed to implement the algorithm. We updated the sentence to clarify it as follows: 

“In computer science, algorithms are typically implemented in software, which consists of code executed on processor-based hardware. However, they can also be realized in programmable hardware (e.g. FPGAs) or a combination of both.”

Comment 4: The implementation of ROS is very critical. In case there is no real robot, all courses can be run, except for the electronics and embedded systems courses.

Response 4: The implementation of the Robot Operating System (ROS) holds significant importance in our approach and is among our next plans to integrate its support. Indeed most courses can be effectively run without a physical robot for a range of topics, and we are working on the development of a digital twin of FOSSbot that can support this option. However, as you notice, practical hands-on experience is particularly crucial for electronics and embedded systems courses. These courses heavily rely on the direct interaction with hardware components, and the absence of a physical robot could limit the practical application of theoretical concepts in these specific domains. Overall the use of a robot as the common platform for teaching all courses adds to the homogeneity of the approach and gives a practical aspect on the educational process.

We added the following in the conclusions:

“In the absence of a physical robot, virtual simulations through ROS provide an alternative avenue for students to explore and experiment with various aspects of robotics, fostering a deeper understanding of algorithms and software implementation.”

Reviewer 3 Report

Comments and Suggestions for Authors

Dear Authors,

Thank you for your article titled "A Holistic Approach to Use Educational Robots in the Development of IT Curricula." I found your work to be informative and relevant to the field of computer science education. I have a few comments and suggestions that I believe could further enhance your article:

Abstract: The abstract provides a clear overview of the article. However, it would be beneficial to include a brief statement about the significance of incorporating educational robots in higher education curricula. This would help readers understand the importance of your research.

Introduction: The introduction effectively introduces the topic of educational robots and their role in K-12 education. To provide a comprehensive overview, you may consider including a brief discussion on the current state of computer science education at the college and university level and how educational robots can address any existing challenges or limitations.

Literature Review: While you mention a significant gap in the literature regarding a curriculum fully integrated with an educational robot, it would be helpful to provide a more detailed analysis of the existing literature on educational robots in higher education. This would strengthen the novelty and contribution of your proposed curriculum.

Proposed Curriculum: The description of the proposed one-year computer science curriculum is well-presented. However, it would be valuable to provide some insights into the expected learning outcomes or competencies that students would acquire through each course. Additionally, discussing the potential career pathways or industry relevance of the skills developed in this curriculum would be beneficial.

FOSSBot: The selection of FOSSBot as the educational robot for the curriculum is justified, and its features are well-explained. To further support your choice, you could include a brief comparison with other available educational robots in terms of cost, functionality, and customizability.

Practical Activities: The inclusion of hands-on activities throughout the curriculum is commendable. It would be helpful to provide some examples or case studies of specific activities that students could engage in for each course. This would give readers a clearer understanding of the practical implementation of the curriculum.

Conclusion: The conclusion effectively summarizes the main points of the article. However, I would suggest adding a section on future directions or potential areas of further research. This would encourage readers and researchers to explore the topic beyond the scope of your current work.

Overall, I believe your article makes a valuable contribution to the field of computer science education. By addressing the suggested points, you can further strengthen the clarity and impact of your research. Thank you for your efforts, and I wish you the best in your future endeavors.

Comments on the Quality of English Language

Based on the provided excerpt from the article, the quality of English language appears to be generally good. The sentences are coherent, and the ideas are clearly expressed. The language used is technical and academic in nature, which is appropriate for an article in the field of computer science and education. There are no glaring grammatical errors or inconsistencies in the language usage. Overall, the English language quality in the excerpt is of a satisfactory standard for a scholarly publication.

Author Response

We would like to thank the reviewer for the detailed comments and suggestions. We incorporated everything in this revised manuscript, using blue font to indicate the changes.

Comment 1: Thank you for your article titled "A Holistic Approach to Use Educational Robots in the Development of IT Curricula." I found your work to be informative and relevant to the field of computer science education. I have a few comments and suggestions that I believe could further enhance your article:  

Abstract: The abstract provides a clear overview of the article. However, it would be beneficial to include a brief statement about the significance of incorporating educational robots in higher education curricula. This would help readers understand the importance of your research.

Response 1:  Thank you for your positive comment. We added the following sentence at the end of the abstract:

“With our work, we show that by incorporating educational robots in higher education we can address this gap and provide a new ledger for boosting tertiary education.”

Comment 2: Introduction: The introduction effectively introduces the topic of educational robots and their role in K-12 education. To provide a comprehensive overview, you may consider including a brief discussion on the current state of computer science education at the college and university level and how educational robots can address any existing challenges or limitations.

Response 2:  To cover the point that you identified, we added one more paragraph (and a couple of citations) to emphasize the current situation in research concerning the use of educational robots in tertiary education. This paragraph introduces the need for a holistic approach and explains why an open-source robot can be a viable solution. The paragraph that we added is as follows:

The existing literature on educational robotics in tertiary education mostly focuses on using commercial robots or their virtual simulations for supporting specific courses and teaching specific subjects each time [39 ,40]. They usually lack a holistic approach in using the same robot to support all the courses of a curriculum and provide activities that fit the needs of each specific course. The main objective of the current work is to fill this gap by using an open-source robot and providing an approach for using it in multiple courses of a Computer Science curriculum. We believe that an open-source robot allows for more customization and expansions and can be the common basis for a holistic approach in the use of Educational Robots for supporting multiple courses of a curriculum. This provides better integration of emerging technologies -especially robotics- into higher education through actual practice and engagement [41].

The references we added follow:

“39. Spolaôr, N.; Benitti, F. B. V. Robotics applications grounded in learning theories on tertiary education: A systematic review. In Computers and Education, 2017, 112, 97-107.

  1. Atman Uslu, N.; Yavuz, G. Ö.; Kocak Usluel, Y. A systematic review study on educational robotics and robots. Interactive Learning Environments, 2023, 31(9), 5874-5898.
  2. Leoste, J.,; Jõgi, L.; Õun, T.; Pastor, L.; San Martín López, J.; Grauberg, I. Perceptions about the future of integrating emerging technologies into higher education—the case of robotics with artificial intelligence. Computers, 2021, 10(9), 110.”

Comment 3: Literature Review: While you mention a significant gap in the literature regarding a curriculum fully integrated with an educational robot, it would be helpful to provide a more detailed analysis of the existing literature on educational robots in higher education. This would strengthen the novelty and contribution of your proposed curriculum.

Response 3:  Thank you for this comment. See our response to the previous comment. In addition, Table 1 in the Literature Review section lists several educational robots that are used in various educational levels, some of them in tertiary education too. The last two paragraphs in section 2 explain this sporadic use of educational robots in different courses and the lack of a holistic approach.

Comment 4: Proposed Curriculum: The description of the proposed one-year computer science curriculum is well-presented. However, it would be valuable to provide some insights into the expected learning outcomes or competencies that students would acquire through each course. Additionally, discussing the potential career pathways or industry relevance of the skills developed in this curriculum would be beneficial.

Response 4:  Thank you for this constructive comment. Following your advice we added this information at the end of each course description.

Comment 5: FOSSBot: The selection of FOSSBot as the educational robot for the curriculum is justified, and its features are well-explained. To further support your choice, you could include a brief comparison with other available educational robots in terms of cost, functionality, and customizability.

Response 5:   Thank you for this comment, which allows us to further justify our choice. We added the following paragraph at the end of section 3:

“Compared to other commercial and research robots that are used in educational robotics, as listed in Table 1 and analyzed in [42], FOSSbot offers a rich inventory of sensors at a low cost, nearly 100 euros, which can be further reduced if the battery recharge module is omitted. It is not as popular as LEGO or other commercial robots, but it is as flexible as most research robots, and this flexibility is enhanced by its open-source and open-design nature. This flexibility makes it ideal for a wide range of courses [42]. “

We also added a new citation:

“42. Kalaitzidou, M.; Pachidis, T. P. Recent Robots in STEAM Education. Education Sciences, 2023. 13(3), 272.”

Comment 6: Practical Activities: The inclusion of hands-on activities throughout the curriculum is commendable. It would be helpful to provide some examples or case studies of specific activities that students could engage in for each course. This would give readers a clearer understanding of the practical implementation of the curriculum.

Response 6: We have already included activities linked to the courses and to the robot sensors in Tables 2 and 3. However, following your suggestions, we added a few paragraphs in the respective subsection to provide some indicative activity examples. The newly added paragraphs are as follows:

The teaching of courses in the curriculum can be enriched with hands-on activities that exemplify the practical implementation of each course. In the Electronics course, students wire the Raspberry Pi microprocessor with sensors and actuators, gaining practical experience in working with electronic components like transistors, capacitors, resistors, and diodes. They further enhance their skills by programming the microcontroller to execute specific tasks. In the Embedded Systems course, students orchestrate the operation of sensors and actuators using the GPIO pins on the Raspberry Pi, providing a tangible understanding of embedded systems concepts.

Moving to the Python Programming course, students can learn basic programming skills by utilizing the core robot class and writing functions for specific tasks. The Algorithms course focuses on implementing navigation algorithms based on graph theory, showcasing how students provide instructions to control the robot's movements and find the shortest path. In the Data Science course, learners use sensors to collect data and apply data acquisition techniques, gaining hands-on experience in visualizing data through plots. This practical application reinforces their understanding of data science concepts.

In the Artificial Intelligence course, the integration of a microphone and camera allows students to interact with the robot using voice commands and perform object detection and avoidance. This real-world application demonstrates how AI models can be applied to enhance the robot's capabilities and interaction with users. The curriculum further extends to all other courses integrating specific libraries and activities tailored to the respective domain for a comprehensive and hands-on learning experience. This emphasis on practical examples across various courses ensures that students develop tangible skills applicable to real-world scenarios.”

Comment 7: Conclusion: The conclusion effectively summarizes the main points of the article. However, I would suggest adding a section on future directions or potential areas of further research. This would encourage readers and researchers to explore the topic beyond the scope of your current work.

Response 7:  Thank you for this comment. We already had a paragraph with next steps in the Conclusions section. We enhanced this part to be more descriptive of our future research plans.

Comment 8: Overall, I believe your article makes a valuable contribution to the field of computer science education. By addressing the suggested points, you can further strengthen the clarity and impact of your research. Thank you for your efforts, and I wish you the best in your future endeavors.

Response 8:  Thank you for your positive and valuable feedback. We hope that with the new additions we successfully addressed the suggested points and improved our manuscript.

Comment 9: Comments on the Quality of English Language

Based on the provided excerpt from the article, the quality of English language appears to be generally good. The sentences are coherent, and the ideas are clearly expressed. The language used is technical and academic in nature, which is appropriate for an article in the field of computer science and education. There are no glaring grammatical errors or inconsistencies in the language usage. Overall, the English language quality in the excerpt is of a satisfactory standard for a scholarly publication.

Response 9:  Thank you for your positive comment.

Round 2

Reviewer 3 Report

Comments and Suggestions for Authors

Strengths:

Clear Objective: The article clearly states its objective of proposing a holistic approach to using educational robots in tertiary education. The objective is well-defined and specific.

Comprehensive Literature Review: The article includes a literature review that discusses the existing use of educational robots in different educational levels and highlights the gap in incorporating educational robots in tertiary education. This provides a context for the proposed approach.

Practical Application: The article proposes the use of an open-source educational robot, FOSSBot, and describes its capabilities and potential applications in various computer science courses. The practical implementation of the proposed approach adds value to the article.

Curriculum Design: The article presents a one-year computer science curriculum that integrates the use of educational robots. It outlines the courses and their practical components, showing how the robot can be utilized in each course. This provides a structured framework for implementation.

Weaknesses:

Lack of Empirical Data: The article does not provide empirical data or results from implementing the proposed approach. While it discusses the potential benefits and applications of educational robots, the lack of empirical evidence limits the strength of the argument.

Limited Discussion on Pedagogical Impact: The article focuses more on the technical aspects of using educational robots and their functionalities. It would benefit from a deeper discussion on the pedagogical impact of incorporating educational robots in tertiary education, including the potential benefits for student learning outcomes.

Limited Discussion on Challenges: The article briefly mentions the limitations of using commercial robots and the need for a holistic approach, but it does not extensively discuss the challenges or potential drawbacks of implementing educational robots in tertiary education. A more thorough analysis of the challenges and potential solutions would enhance the article's credibility.

Suggestions for Improvement:

Empirical Evaluation: Conduct empirical studies or provide examples of real-world implementations to validate the proposed approach. This could include gathering data on student performance, engagement, and satisfaction when using educational robots in specific courses.

Pedagogical Considerations: Expand the discussion on the pedagogical impact of educational robots in tertiary education. Consider exploring how the use of educational robots enhances student engagement, problem-solving skills, critical thinking, and collaborative learning.

Address Challenges: Provide a more in-depth analysis of the challenges that may arise when implementing educational robots in tertiary education. Discuss potential solutions or strategies to overcome these challenges, such as training faculty, addressing resource constraints, and ensuring inclusivity.

Comparative Analysis: Compare the proposed approach with existing approaches that use commercial robots or virtual simulations. Highlight the advantages and unique features of the proposed approach and how it addresses the limitations of other approaches.

Analyze more, more relevant and newer articles: ---Load-balanced and QoS-aware software-defined Internet of Things. ---Software-defined Internet of Multimedia Things: Energy-efficient and Load-balanced Resource Management.

Comments on the Quality of English Language

Grammar and Syntax: The paper generally demonstrates a good command of grammar and syntax. Sentences are structured correctly, and there are minimal grammatical errors. However, occasional errors, such as missing articles or subject-verb agreement issues, can be found. A thorough proofreading would help address these minor errors.

Vocabulary and Word Choice: The paper utilizes a suitable academic vocabulary and employs technical terms relevant to the field of study. The choice of words generally conveys the intended meaning effectively. However, there is room for improvement in terms of using more precise and concise language in some instances.

Clarity and Coherence: The paper maintains a clear and coherent presentation of ideas. The sentences and paragraphs flow logically, allowing readers to follow the arguments and understand the content. The use of appropriate transitional phrases and cohesive devices enhances the overall clarity.

Academic Style: The paper adheres to an appropriate academic style, employing formal language and maintaining an objective tone. It effectively uses citations to support claims and acknowledge existing literature. However, there could be some areas where further elaboration or clarification is needed to enhance the paper's academic rigor.

Sentence Structure and Variety: The paper demonstrates a good range of sentence structures, including simple, compound, and complex sentences. This variety contributes to the overall readability and engagement of the content.

Technical Terminology: The paper effectively incorporates technical terminology related to educational robots and computer science. This demonstrates a strong understanding of the subject matter and helps to convey ideas accurately within the field.

Proofreading: While the language quality is generally good, a thorough proofreading process would be beneficial to identify and rectify any remaining minor errors, such as punctuation inconsistencies or typographical mistakes.

Author Response

We would like to thank you for the additional comments that helped us improve the manuscript. 

Given the time, tools, and methods restrictions, we did our best to address the concerns that were raised. In the following, we outline how the comments have been taken into account and have been incorporated into this second revision of the article. To this end, we use blue-colored fonts in the document to mark any -major updates.

-----

Comment 1: Lack of Empirical Data: The article does not provide empirical data or results from implementing the proposed approach. While it discusses the potential benefits and applications of educational robots, the lack of empirical evidence limits the strength of the argument.

Response 1: Thank you for this comment. Following your suggestion, we added a new section (section 5) that describes the evaluation process that we followed so far. The approach is two-folded: first evaluates the performance of students in an IoT course at the university, and second, it records the experiences of students who practiced various computer science topics using the robot in a less formal educational context.

Comment 2: Limited Discussion on Pedagogical Impact: The article focuses more on the technical aspects of using educational robots and their functionalities. It would benefit from a deeper discussion on the pedagogical impact of incorporating educational robots in tertiary education, including the potential benefits for student learning outcomes.

Response 2: Thank you for this constructive comment. We believe that the discussion of the pedagogical impact of the use of an educational robot in multiple courses in tandem could be the subject of a whole new research work. However, following your advice, we added two paragraphs at the end of the evaluation section 5 that further discuss the impact our approach can have in tertiary education, from the pedagogical perspective. 

Comment 3: Limited Discussion on Challenges: The article briefly mentions the limitations of using commercial robots and the need for a holistic approach, but it does not extensively discuss the challenges or potential drawbacks of implementing educational robots in tertiary education. A more thorough analysis of the challenges and potential solutions would enhance the article's credibility.

Response 3: Thank you for your comment, which allows us to perform a better analysis of the impact of our proposed solution. Although some of the challenges were already listed at the end of section 2, we added an extra paragraph in section 2 that highlights the main challenges that arise from the use of educational robots in tertiary education and gives examples of potential solutions.

Suggestions for Improvement:

Comment 4:  Empirical Evaluation: Conduct empirical studies or provide examples of real-world implementations to validate the proposed approach. This could include gathering data on student performance, engagement, and satisfaction when using educational robots in specific courses.

Response 4: Thank you for this suggestion. See our response to comment 1.

Comment 5: Pedagogical Considerations: Expand the discussion on the pedagogical impact of educational robots in tertiary education. Consider exploring how the use of educational robots enhances student engagement, problem-solving skills, critical thinking, and collaborative learning.

Response 5: Thank you for this suggestion. See our response to comment 2.

Comment 6: Address Challenges: Provide a more in-depth analysis of the challenges that may arise when implementing educational robots in tertiary education. Discuss potential solutions or strategies to overcome these challenges, such as training faculty, addressing resource constraints, and ensuring inclusivity.

Response 6: Thank you for this suggestion. See our response to comment 3.

Comment 7: Comparative Analysis: Compare the proposed approach with existing approaches that use commercial robots or virtual simulations. Highlight the advantages and unique features of the proposed approach and how it addresses the limitations of other approaches.

Response 7: Thank you for this comment. Section 2 provides a brief evaluation of existing approaches and compares them with the proposed approach. The main advantages rely on: i) the use of an open-source robot, which can be extended to cover varying needs, ii) the low cost of the solution in combination with a wide range of sensors. These two allow for developing solutions that perfectly fit to the needs of the various educational scenarios in each course. We expanded the respective paragraph to provide more information on the advantages of our solution.

Comment 8: Analyze more, more relevant and newer articles: ---Load-balanced and QoS-aware software-defined Internet of Things. ---Software-defined Internet of Multimedia Things: Energy-efficient and Load-balanced Resource Management.

Response 8: We would like to thank the reviewer for this comment. The two articles mostly focus on energy efficiency and load-balanced management of IoT resources. Thus they do not directly related to the concept of this article. 20 out of the 42 references in this article are within the last 5 years, so we believe that the cited articles are relevant and new.

Comments on the Quality of English Language

Comment 9: Grammar and Syntax: The paper generally demonstrates a good command of grammar and syntax. Sentences are structured correctly, and there are minimal grammatical errors. However, occasional errors, such as missing articles or subject-verb agreement issues, can be found. A thorough proofreading would help address these minor errors.

Response 9: Thank you for this comment. We carefully re-read the article and corrected minor errors. We used Grammarly for double checking. 

Comment 10: Vocabulary and Word Choice: The paper utilizes a suitable academic vocabulary and employs technical terms relevant to the field of study. The choice of words generally conveys the intended meaning effectively. However, there is room for improvement in terms of using more precise and concise language in some instances.

Response 10: Thank you for this positive comment. After careful proofreading, we performed minor language corrections.

Comment 11: Clarity and Coherence: The paper maintains a clear and coherent presentation of ideas. The sentences and paragraphs flow logically, allowing readers to follow the arguments and understand the content. The use of appropriate transitional phrases and cohesive devices enhances the overall clarity.

 Response 11: Thank you for this positive comment. 

Comment 12: Academic Style: The paper adheres to an appropriate academic style, employing formal language and maintaining an objective tone. It effectively uses citations to support claims and acknowledge existing literature. However, there could be some areas where further elaboration or clarification is needed to enhance the paper's academic rigor.

Response 12:  Thank you for this comment. In the first revision, we made some clarifications to enhance the academic style of the paper, following the reviewers’ suggestions.

Comment 13: Sentence Structure and Variety: The paper demonstrates a good range of sentence structures, including simple, compound, and complex sentences. This variety contributes to the overall readability and engagement of the content. 

Technical Terminology: The paper effectively incorporates technical terminology related to educational robots and computer science. This demonstrates a strong understanding of the subject matter and helps to convey ideas accurately within the field.

Response 13: Thank you for your positive comments.

Comment 14: Proofreading: While the language quality is generally good, a thorough proofreading process would be beneficial to identify and rectify any remaining minor errors, such as punctuation inconsistencies or typographical mistakes.

Response 14: Thank you for this comment. See our response in comment 10.

Round 3

Reviewer 3 Report

Comments and Suggestions for Authors

The article discusses the integration of educational robots, specifically FOSSBot, into computer science courses to enhance learning experiences. In this context, referring to the following references would provide logical reasons and technical justifications for the necessity of incorporating load-balanced and quality of service (QoS)-aware software-defined Internet of Things (IoT) and software-defined Internet of Multimedia Things into the discussion:

  1. Load-balanced and QoS-aware software-defined Internet of Things:
  • Logical Reason: This reference would offer insights into how load balancing and QoS considerations are crucial in IoT environments, especially when dealing with a network of interconnected devices like educational robots. Understanding how to manage resources efficiently and ensure quality service delivery is essential for optimizing the performance of IoT systems.
  • Technical Justification: By referencing this work, the article can delve into the technical aspects of implementing load balancing and QoS mechanisms in IoT settings. This would provide a deeper understanding of how these concepts can be applied to educational robots like FOSSBot to improve their functionality, network performance, and overall user experience.
  1. Software-defined Internet of Multimedia Things: Energy-efficient and Load-balanced Resource Management:
  • Logical Reason: This reference would contribute valuable insights into managing multimedia data in IoT environments, which is relevant when educational robots like FOSSBot are equipped with sensors, cameras, and other multimedia capabilities. Energy efficiency and load balancing are critical factors in optimizing resource utilization in such scenarios.
  • Technical Justification: By referencing this work, the article can explore the technical aspects of energy-efficient resource management and load balancing in multimedia-rich IoT environments. This would provide a solid foundation for understanding how to optimize resource usage, enhance performance, and ensure efficient operation of educational robots with multimedia capabilities.

Incorporating these references would not only strengthen the theoretical framework of the article but also provide practical guidance on implementing advanced networking and resource management techniques in the context of educational robotics and computer science education.

Author Response

Thank you for the suggested reasoning and justification. We included the relevant content in this revised revision and used the respective references.

More specifically in the Computer Vision section 4.1.8 we added the following:

“Running computer vision tasks on an educational robot would contribute valuable insights into managing multimedia data in the edge, and considering energy efficiency and load balancing which are critical factors in optimizing resource utilization in such scenarios. Students can explore the technical aspects of energy-efficient resource management and load balancing in multimedia-rich robotics and will get a solid foundation for understanding how to optimize resource usage, enhance performance, and ensure the efficient operation of educational robots with multimedia capabilities [43].”

And in the IoT section 4.1.11 we added the following:

“Another important aspect of IoT that can be examined using the robot is load balancing and QoS. Understanding how to manage resources efficiently and ensure quality service delivery is essential for optimizing the performance of IoT systems especially when dealing with a network of interconnected devices like educational robots. Students can delve into the technical aspects of implementing load balancing and QoS mechanisms in IoT settings and get a deeper understanding of how these concepts can be applied to improve a robot's functionality, network performance, and overall user experience [44].”

where:

  1. Montazerolghaem, A. Software-defined Internet of Multimedia Things: Energy-efficient and Load-balanced Resource Management. IEEE Internet of Things Journal, 2021. 9(3), 2432-2442. 
  2. Montazerolghaem, A.; Yaghmaee, M. H. Load-balanced and QoS-aware software-defined Internet of Things. IEEE Internet of Things Journal, 2020. 7(4), 3323-3337 
